# Effect of Different Drying Techniques on Total Bioactive Compounds and Individual Phenolic Composition in Goji Berries

**Busra Turan** [1], **Zeynep Hazal Tekin-Cakmak** [1,2], **Selma Kayacan Çakmakoglu** [1], **Salih Karasu** [1,*], **Muhammed Zahid Kasapoglu** [3] **and Esra Avci** [1,4]

1    Department of Food Engineering, Faculty of Chemical and Metallurgical Engineering,
     Yildiz Technical University, Istanbul 34210, Turkey
2    Department of Nutrition and Dietetics, Faculty of Health Sciences, Istinye University, Istanbul 34010, Turkey
3    Istanbul Teknokent, Istanbul University Cerrahpasa Avcılar Campus, Istanbul 34320, Turkey
4    Bypro Functional Food and Biotechnology, Istanbul 34210, Turkey
*    Correspondence: skarasu@yildiz.edu.tr

**Abstract:** In recent years, interest in the consumption of dried goji berries has increased due to its high bioactive properties. Alternative drying methods that provide faster drying and better preservation of bioactive properties should be developed. This study aims to investigate the effect of different drying methods on the drying time and quality characteristics of the goji berry; namely, hot-air drying (HAD), ultrasound-assisted vacuum drying (USVD), vacuum drying (VD), freeze-drying (FD), and ultrasound-pretreated freeze-drying (USFD). The drying kinetic, total phenolic content, antioxidant capacity, phenolic profile, carotenoid profile, and color change of the goji berry fruit were determined. The drying times for VD, USVD, and HAD varied between 275–1330 min. USVD treatment showed faster drying behavior than the other two drying methods. The total phenolic content (TPC) value of dry samples varied between 1002.53–1238.59 mg GAE/g DM. USVD treatments exhibited a higher total phenolic content (TPC) value than all other drying methods. DPPH and CUPRAC values varied between 15.70–29.90 mg TE/g DM and 40.98–226.09 mg TE/g DM, respectively. The total color change (ΔE) value ranged between 4.59 and 23.93 and HAD dried samples showed the highest ΔE of all samples. The results of the phenolic profile were consistent with TPC analysis. HAD caused higher phenolic compound degradation than VD, USVD, and FD. The results of this study showed that different drying techniques significantly affected the drying rate and retention of bioactive components of the goji berry fruit, and the USVD and VD methods could be used as an alternative to the HAD method. This study concluded that USVD and FD could be considered as suitable drying methods and could be used as alternatives to HAD in the drying of goji berries.

**Keywords:** goji berry; drying; phenolic compounds; ultrasound-assisted vacuum drying; ultrasound-assisted freeze-drying

## 1. Introduction

The goji berry (*Lycium barbarum* L.), also known as Chinese wolfberry and *Lycium* fruit is a member of the Solanaceae family [1]. It grows in China, Tibet, and other parts of Asia. It has been reported that goji berries have several bioactive properties, such as antioxidant, immunomodulatory, lipid lowering, hypoglycemic, and anticancer properties based on the findings of numerous studies. As a result, including goji berries in a balanced diet could prevent many aging-related disorders. The polysaccharides, carotenoids, polyphenols, and other bioactive components of goji berries are thought to be responsible for these health-promoting properties [2].

Goji berries are susceptible to chemical alterations and microbial spoilage after harvesting due to their high moisture and bioactive content [1]. Goji berries have a short fruiting

period and must be exposed to a preservation method for all-year-round consumption [3]. Drying has emerged as one of the key methods for processing goji berries due to the resulting increased shelf life and expanded distribution network. It also meets the needs of space-saving industrial production and high productivity [1]. Hot-air drying is the most popular technique because it is simple to use. However, long drying periods and high temperatures cause negative effects on microstructural properties and the degradation of bioactive compounds [4,5]. Instead of conventional methods, alternative drying methods that provide faster drying and better preservation of bioactive components should be developed. The freeze-drying process, which is regarded as the best drying method, is favorable in terms of achieving dried products with excellent sensory and nutritional quality as well as a high rehydration capability [6]. However, the high cost of the drying process and the long drying time limit the use of FD drying in many products. The alternative method is vacuum drying, which is frequently used in place of hot-air drying to enhance the quality of dried goods by reducing drying temperatures or drying times. Due to water evaporation at temperatures lower than atmospheric pressure, fruits are dried using the vacuum drying process without being subjected to high temperatures. Furthermore, because dehydration occurs in an oxygen-free environment, it lowers oxidation processes. These benefits increase the physical quality of dried items such as color, taste, and scent [7,8]. Several studies have been published on the use of ultrasound for pre-treatment before drying to enhance drying kinetics and lower energy expenditures [9]. The moisture transfer resistance during the drying process is the most important problem affecting the drying rate, and new approaches are needed to reduce moisture transfer resistance. Ultrasound application can be considered as one of these approaches. Thanks to the ability of the ultrasound waves to squeeze and release the water repeatedly, microscopic cavities are formed in the tissues of fruits and vegetables, so that the bound water attached to the tissues and free water could be removed. Microscopic cavities improve diffusion, increase mass transfer, and reduce the drying time [10]. Ultrasound application can be a good alternative to other drying methods for reducing the drying time and resulting in less changes to bioactive components and color properties. There are many studies in which fruits and vegetables are dried with the assistance of ultrasound. Some of these are green beans [11], red peppers [10], pomegranate arils [5], jackfruit [12], strawberry slices [13], pineapple slices [14], raspberry [15], and Asian pear [16]. Studies on drying goji berry fruits by combining osmotic dehydration and air-drying [17], sodium carbonate pretreatment and hot-air drying [18], ultrasonic pretreatment with an electrohydrodynamic drying process [19], radio frequency-hot air combined drying [14] were conducted. However, there was no attempt to use USVD and ultrasound-pretreated freeze-drying (USFD).

In this study, HAD, VD, USVD, FD, and USFD were used to dry goji berries and the effects of the drying methods on drying kinetics, TPC, antioxidant capacity, phenolic composition and color properties of the goji berry were investigated.

## 2. Materials and Methods

### 2.1. Materials

Goji berry (*Lycium barbarum*) was harvested in September 2021 from a local producer (Ayşepınar Ecological Agricultural Farm) in Manisa, Akhisar region, Turkey. The fresh goji berry samples were transported to the laboratory and kept at 4 °C until the drying process began. The moisture contents of fresh goji sample were found to be 73.70%. Goji berries were dried as a whole and stored in a desiccator until the analysis. All chemicals (HPLC grade-methanol, -ethanol, -acetonitrile, -ultra-distilled water, and acetic acid, hexane, NaCl, KOH, Folin–Ciocalteu's phenol reagent, $Na_2CO_3$, 1,1-diphenyl-2-picrylhydrazyl (DPPH), $CuCl_2$, neocuproine, and $NH_4Ac$) and standards were analytical grade and bought from Merck (Darmstadt, Germany) or Sigma-Aldrich (Sigma Chemical Co., St. Louis, MO, USA).

*2.2. Methods*

2.2.1. Drying Procedure

HAD, VD, USVD, FD, and USFD drying methods were used to dry goji berries. According to the preliminary investigation, the drying temperature was chosen as 50 °C for HAD, VD, and USVD based on the retention of the bioactive compounds. The dried sample was evaluated by comparing to the fresh samples. The lower change in color and bioactive compounds, as well as the shorter drying time, were considered success criteria for the methods. HAD was conducted at a constant flow rate of 1.3 m/s. Testo 440 vane probe anemometer (Lutron, AM-4201, Taipei, Taiwan) was utilized for the air velocity measurement. The goji berries' surface was exposed to the airflow in a horizontal direction. For VD, a vacuum drier (DaihanWOV-30, Gangwon-do, Republic of Korea) was used for vacuum drying goji berries. A vacuum pump (EVP 2XZ-2C, Zhejiang, China) with a 60 mbar ultimate pressure and a 2 L/s pump speed was used to control the vacuum. For the USVD method, goji berries were held at 30 min ultrasound water bath (Daihan, WUC-D10H, Gangwon-do, Republic of Korea) (amplitude: 100%, power intensity: ~1 W/cm$^2$, volume: 10 L) and then dried in a vacuum drier (DaihanWOV-30, Gangwon-do, Republic of Korea). An ultrasonic water bath was used to sonicate at a frequency of 40 kHz. FD was carried out in accordance with a protocol for a freeze dryer (Martin Christ, Beta 1–8 LSC plus, Osterode am Harz, Germany). The fresh goji berries were frozen for 24 h at −80 °C before the freeze-drying process. The weight loss of goji berries was measured at 30 min intervals during HAD, VD, and USVD. Goji berries were dried until their moisture level reached 0.2 kg water/kg dry matter before the drying process was stopped.

2.2.2. The Modeling of the Dehydration Characteristic and Effective Moisture Diffusivity ($D_{eff}$)

Table 1 shows nine mathematical models simulating the drying time and moisture data.

**Table 1.** Thin layer drying kinetic models used to evaluate the data obtained from drying goji berry samples with different methods.

| Model | Equation | References |
|---|---|---|
| Newton | $Y = exp(-kt)$ | Lewis [20] |
| Page | $Y = exp(-kt^n)$ | Page [21] |
| Henderson ve Pabis | $Y = aexp(-kt^n)$ | Henderson and Pabis [22] |
| Logarithmic | $Y = aexp(-kt^n) + c$ | Yağcıoğlu, Değirmencioğlu [23] |
| Wang and Singh | $Y = 1 + at + bt^2$ | Wang and Singh [24] |
| Two terms | $Y = aexp(-k_0t) + bexp(-k_1t)$ | Zielinska and Markowski [25] |
| Two terms exponential | $Y = aexp(-kt) + (1-a)exp(-kat)$ | Sharaf-Eldeen, Blaisdell [26] |
| Parabolic | $Y = a + bt + ct^2$ | Sharma and Prasad [27] |
| Weibull | $Y = exp(-(t/b)^a)$ | Corzo, Bracho [28] |

Y: moisture content; a, b, c, and n: the drying coefficients; k, $k_0$ and $k_1$: the drying constants.

Because $M_e$ is very modest in comparison to M or $M_0$, the moisture ratio was simplified to $M/M_0$ rather than $(M - M_e)/(M_0 - M_e)$, where M reflects the moisture content at time t, $M_0$ demonstrates the beginning moisture content, and $M_e$ represents the equilibrium level of moisture. The STATISTICA software package and nonlinear regression were employed to generate the parameters of the models and $R^2$ values (StatSoft, Tulsa, OK, USA). The $R^2$ and root mean square error (RMSE) values were used to assess the model's success; lower RMSE and higher $R^2$ values indicated the model's applicability. The following equation was used to calculate the RMSE values [29]:

$$\text{RMSE} = \left[ \frac{1}{N} \sum_{i=1}^{N} \left( \text{MR}_{\text{pre,i}} - \text{MR}_{\text{exp,i}} \right)^2 \right]^{1/2} \tag{1}$$

In Equation (1), $\text{MR}_{\text{prei}}$ reflects the predicted moisture ratio, $\text{MR}_{\text{expi}}$ shows the experimental moisture ratio, N reflects the number of observations, and n indicates the number of constants.

Fick's second law was utilized to calculate of the $D_{\text{eff}}$ value of goji berries (Equation (2)):

$$\frac{\partial M}{\partial t} = \nabla [D_{\text{eff}} (\nabla M)] \tag{2}$$

The following equation, which takes into account spherical forms and unstable diffusion circumstances, was adapted using this equation:

$$\text{MR} = \frac{6}{\pi^2} \sum_{n=1}^{\infty} \frac{1}{n^2} \exp \left( -n^2 \pi^2 \frac{D_{\text{eff}} t}{r^2} \right) \tag{3}$$

where r is the goji berry's half-thickness (m), $D_{\text{eff}}$ reflects the effective moisture diffusivity ($m^2$/s), and n is a positive integer. This equation can be simplified as Equation (4) for lengthy drying times:

$$\ln(\text{MR}) = \ln \left( \frac{6}{\pi^2} \right) - \left( \pi^2 \frac{D_{\text{eff}} t}{r^2} \right) \tag{4}$$

2.2.3. Extraction Procedure

Fresh and dried goji berries were mixed with a 1:10 *w/v* ratio of methanol-water (50:50, *v/v*) solution. Using an Ultra-Turrax, the goji berry samples and solution were homogenized at 10,000 rpm for 2 min (Daihan, HG-15D, Gangwon-do, Korea). The mixture was then shaken for 2 h at 25 degrees Celsius. The mixture was stirred and thereafter centrifugation was conducted at $2800\times$ *g* for 10 min. Finally, a 0.45 m syringe filter was used to filtrate the supernatant. [30].

2.2.4. Determination of Total Phenolic Content (TPC) and Antioxidant Capacity

The TPC value was determined according to a modified method reported by Singleton and Rossi [31]. Folin–Ciocalteu's phenol reagent (1:10 *v/v*) and $Na_2CO_3$ (7.5 g/100 g) were added to diluted methanolic goji berry extracts (0.5 mL) in amounts of 2.5 mL and 2 mL, respectively. The liquid was then combined in a tube and left in a dark environment for 30 min. A UV-VIS spectrophotometer was used to detect the absorbance at 760 nm (Shimadzu UV-1800, Kyoto, Japan). Gallic acid equivalents (GAE) per gram of dry matter (DM) (mg GAE/g DM) were used to calculate the phenolic content.

The copper-reducing antioxidant capacity (CUPRAC) and 1,1-diphenyl-2-picrylhydrazyl (DPPH) techniques were employed to assess the antioxidant capacity of extracts. The goji berry extract (0.1 mL) was mixed with 4.9 mL DPPH (0.1 mmol/L methanol). At 517 nm, absorbance was measured following a 30-min incubation period in the dark [32]. The CUPRAC test was performed as reported by Apak, Güçlü [33]. For this assay, 1 mL of $CuCl_2$ (10 mmol/L), 1 mL neocuproine (7.5 mmol/L), and 1 mL $NH_4Ac$ (1 mol/L) were mixed; and after that, 1 mL of distilled water and this solution were mixed to bring the total volume up to 4.1 mL. The absorbance values were obtained at 450 nm following an approximately one-hour incubation period in the dark. Trolox equivalents (TE) per gram of dry matter (mg TE/g DM) were used to express the results.

2.2.5. The HPLC Determination of Phenolic Compounds

An HPLC-DAD system was utilized to figure out the amount of each individual phenolic compound (Shimadzu Corp., Kyoto, Japan). The extracts generated using a methanol-based technique to determine the amount of phenolic composition, an HPLC-DAD system (Shimadzu Corp., Kyoto, Japan) was used. A 0.45-m membrane filter was employed to filter extracts from a methanol-water (50:50, *v/v*) solution with a solid-to-liquid ratio of 1:10. A linear gradient eluted program with a mobile phase consisting of solvent A (formic acid/H$_2$O, 0.1:99.9, *v/v*) and solvent B (formic acid/acetonitrile, 0.1:99.9, *v/v*) was used to separate phenolic acids on a Waters Atlantis C18 column (250 mm 4.6 mm, 5 m) at a flow rate of 1 mL/min. The solvent gradient was established in the following manner: 0–15 min linear gradient elution from 10% B to 60% B; 15–20 min isocratic elution from 60% B to 10% B; and 25–30 min isocratic elution from 10% B. By monitoring spectra from 190 to 400 nm, chromatograms were acquired at 278 nm, 320 nm, and 360 nm. To identify phenolic acids, the retention time and absorption spectra of peaks in goji berry samples were compared to those of reference chemicals. Phenolic acids were assessed using calibration curves generated for each of the compounds identified in the samples.

2.2.6. Color

The color analysis of fresh and dried goji berries was carried out using a chromameter (Konica Minolta CR-400, Tokyo, Japan. Color values were defined in terms of *L*\* (whiteness/darkness), *a*\* (red-ness/greenness), and *b*\* (yellowness/blueness), and the total color change (*E*) of the samples was computed using the following equation (Equation (5)):

$$\Delta E = \sqrt{(\Delta L^*)^2 + (\Delta a^*)^2 + (\Delta b^*)^2} \tag{5}$$

Chroma (*C*\*), a quantitative property of colorfulness, was computed using the following equation (Equation (6)):

$$C^* = \sqrt{a^{*2} + b^{*2}} \tag{6}$$

2.2.7. Statistical Analysis

The statistical analysis was completed using the STATISTICA software package (StatSoft, Inc., Tulsa, OK, USA). The relative deviation and mean findings were displayed, and each analysis was carried out in three duplicates. A one-way ANOVA was employed to compare the findings' mean values. Duncan's multiple comparison procedures were used to assess how different drying procedures affected bioactive compounds, phenolic profile changes, and goji berry color. Pearson's correlation coefficient was used to analyze the connection between antioxidant capacity and bioactive substances.

## 3. Results

### 3.1. Drying Kinetics

The images of fresh and dried goji berry samples with different drying methods were given in Figure 1.

The drying curves of goji berry samples by USVD, VD, and HAD methods were shown in Figure 2. As seen in Figure 2, the drying times of the goji berry samples differed according to the methods and were 275, 400, and 1330 min for the USVD, VD and HAD methods, respectively. The moisture content of the goji berry samples gradually decreased with increasing drying time. As seen in Figure 2, the moisture content of the goji berry samples decreased rapidly at the beginning of the drying period. This can be explained by the realization of a constant drying rate for the USV and VD methods.

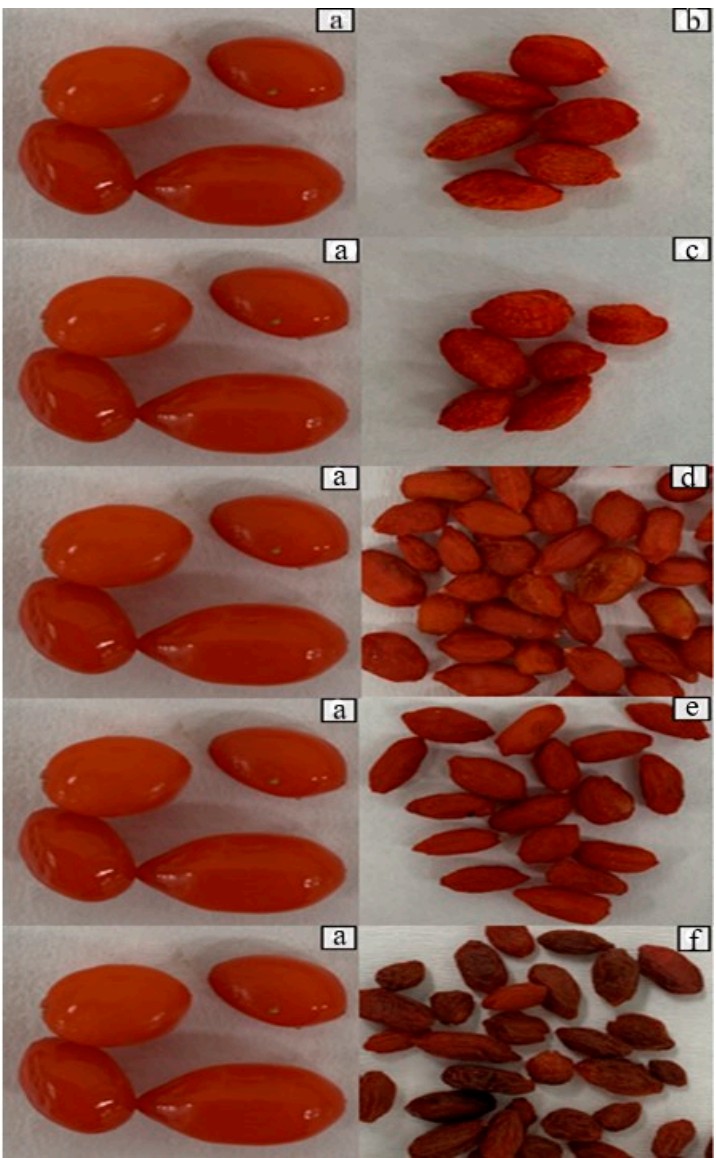

**Figure 1.** Fresh and dried goji berry samples with different drying methods ((**a**): fresh, (**b**): FD, (**c**): USFD, (**d**): VD, (**e**): USVD, (**f**): HAD).

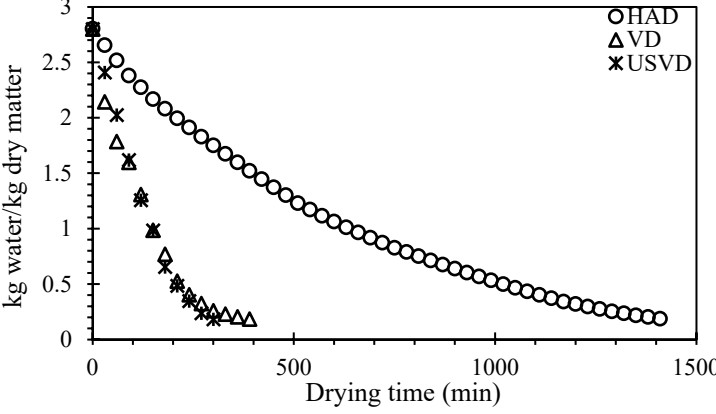

**Figure 2.** Drying kinetics of goji berries (HAD: Hot-air drying; USVD: Ultrasound-assisted vacuum drying; VD: Vacuum drying).

During the hot-air drying process, the slower increase in temperature of the products caused by the external heat flow slows the moisture transfer from the inside, resulting in a longer drying time. During the vacuum drying process, the pressure drops and so does the boiling point of the water contained in the food, resulting in an increased rate of evaporation at the surface. The assisting of ultrasound seems to significantly shorten the drying process. This is so that water can be removed more quickly and the drying process can be sped up because cavitation creates micro gaps in the tissue of fruits and vegetables. Therefore, VD and USVD dry nearly 3.3 and 4.84 times faster than HAD. The lower drying time of USVD compared to HAD and VD was reported in other studies [9,11].

The thin layer drying modeling is useful for optimizing drying conditions, developing and optimizing dryers, and researching complex heat and mass transport processes [34]. For this reason, It is essential to determine the most suitable model representing the drying behavior of the foods. The nine different models are given in Table 1 to describe the drying behavior of samples using HAD, VD, and USVD methods. Statistical analysis values are summarized in Table 2. $R^2$ values for all models are greater than 0.98, showing that all drying models successfully explained the drying behavior of goji berry samples. $R^2$ values ranged between 0.9881 and 0.9995, and RMSE values ranged between 0.031582 and 0.158857. The logarithmic model showed higher $R^2$ and lower RMSE values for the HAD method, Parabolic VD method, and Page and Weibull USVD method compared to other models. Page and Weibull models are other important models for the HAD method ($R^2$:0.999). Batu and Kadakal [35] determined that the Page model is the best model that predicts the lowest experimental moisture content of goji berry berries RMSE and $\chi^2$ and the highest $R^2$ (0.9962–0.9979) for all drying temperatures. The k value and drying kinetic coefficient were used to compare the moisture transfer rates of the samples ($p < 0.05$).

**Table 2.** Parameters and statistical data obtained from different drying models of goji berry fruits dried using HAD, VD, and USVD [1].

| Model | Parameter | Drying Methods | | |
|---|---|---|---|---|
| | | **HAD** | **VD** | **USVD** |
| Newton | k | 0.001656 | 0.007258 | 0.007279 |
| | $R^2$ | 0.9984 | 0.9958 | 0.9883 |
| | RMSE | 0.103547 | 0.098453 | 0.157556 |
| Page | k | 0.001153 | 0.005766 | 0.001377 |
| | n | 1.055739 | 1.045136 | 1.333373 |
| | $R^2$ | 0.9990 | 0.9961 | 0.9995 |
| | RMSE | 0.083415 | 0.095091 | 0.031582 |
| Henderson and Pabis | k | 0.001671 | 0.007244 | 0.007738 |
| | a | 1.008327 | 0.998141 | 1.061663 |
| | $R^2$ | 0.9984 | 0.9958 | 0.9914 |
| | RMSE | 0.101777 | 0.098423 | 0.135938 |
| Logarithmic | k | 0.001371 | 0.006587 | 0.005038 |
| | a | 1.072649 | 1.024806 | 1.271190 |
| | c | −0.089219 | −0.037601 | −0.245337 |
| | $R^2$ | 0.9998 | 0.9963 | 0.9978 |
| | RMSE | 0.032142 | 0.091639 | 0.068074 |
| Wang and Singh | b | 0.00000 | 0.000008 | 0.000008 |
| | a | −0.001317 | −0.005578 | −0.005537 |
| | $R^2$ | 0.9970 | 0.9953 | 0.9989 |
| | RMSE | 0.140594 | 0.103632 | 0.048249 |
| Two-term | a | 0.262317 | 0.125892 | 0.205145 |
| | $k_0$ | 0.001671 | 0.007244 | 0.007739 |
| | $k_1$ | 0.001671 | 0.007244 | 0.007738 |
| | b | 0.746007 | 0.872250 | 0.856508 |

**Table 2.** *Cont.*

| Model | Parameter | Drying Methods | | |
|---|---|---|---|---|
| | | **HAD** | **VD** | **USVD** |
| | $R^2$ | 0.9984 | 0.9958 | 0.9913 |
| | RMSE | 0.101777 | 0.098423 | 0.135938 |
| Two-term exponential | a | 0.001825 | 0.006373 | 0.001905 |
| | k | 0.905164 | 1.130985 | 3.799024 |
| | $R^2$ | 0.9984 | 0.9958 | 0.9881 |
| | RMSE | 0.104389 | 0.09895 | 0.158857 |
| Parabolic | a | 0.9561649 | 0.955698 | 1.019876 |
| | b | −0.001190 | −0.005142 | −0.005788 |
| | c | 0.0000000 | 0.000007 | 0.000009 |
| | $R^2$ | 0.9988 | 0.9971 | 0.9992 |
| | RMSE | 0.088643 | 0.082026 | 0.043587 |
| Weibull | a | 1.0557 | 1.0451 | 1.3334 |
| | b | 606.7118 | 138.8014 | 139.8895 |
| | $R^2$ | 0.9990 | 0.9961 | 0.9995 |
| | RMSE | 0.083415 | 0.095091 | 0.031582 |

[1] HAD: hot-air drying method; USVD: ultrasound-assisted vacuum drying method; VD: vacuum drying method.

The USVD method exhibited the greatest k value for Newton, Henderson, and Pabis, Parabolic, and Weibull drying models followed by VD and HAD methods. These results also showed that the USVD method can significantly increase the drying rate of goji berry samples. Effective moisture diffusion ($D_{eff}$) is a useful indicator of dehydration efficiency [36]. Effective moisture diffusivity for HAD, VD, and USVD at 50 °C was shown in Figure 3. The $D_{eff}$ values obtained as a result of drying the goji berry fruits by HAD, VD, and USVD methods are given in Table 2, and the $D_{eff}$ values were calculated as $3.75 \times 10^{-9}$, $1.36 \times 10^{-8}$, and $1.95 \times 10^{-8}$ m$^2$/s, respectively. The results proved that the $D_{eff}$ values differ according to different drying methods. The highest $D_{eff}$ value is seen in ultrasound-assisted vacuum drying (USVD) due to the cavitation effect. The $D_{eff}$ value was calculated as $1.04 \times 10^{-8}$ m$^2$/s by Batu and Kadakal [35] as a result of drying the goji berry fruits with hot air at 50 °C. Similar results were reported in the Ni, Ding [37] study. The $D_{eff}$ value of the goji berry was reported as $0.421 \times 10^{-9}$–$1.01 \times 10^{-9}$ m$^2$/s. The pretreatment process including the ultrasound application increased the $D_{eff}$ value. Fernandes, Oliveira [38] reported that the ultrasound pretreatment increased the $D_{eff}$ value by more than three times compared with the control. The $D_{eff}$ value of the murta berries for hot-air drying and combined infrared-hot-air drying was reported as $7.59 \times 10^{-10}$–$8.5 \times 10^{-9}$ m$^2$/s [39].

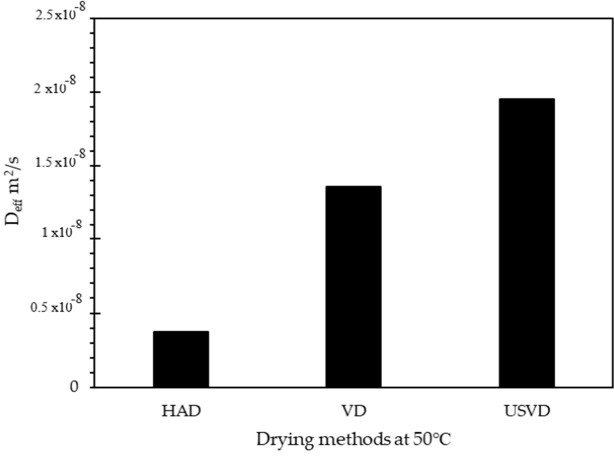

**Figure 3.** Effective moisture diffusivity for HAD, VD, and USVD at 50 °C.

### 3.2. Bioactive Properties of Fresh and Dried Goji Berries

The findings for the TPC level and anti-oxidant potency of fresh and dried goji berries are shown in Table 3. The fresh sample's TPC level was 772.20 mg GAE/g DM. Among the dried goji berry fruits, the highest TPC value belongs to the USVD dried goji berry sample (1238.59 mg GAE/g DM). The TPC result of the samples dried with FD (1115.03 mg GAE/g DM) is close to the samples dried with USVD. The samples dried with FD had a TPC value that is lower than that of the samples dried with VD (1060.54 mg GAE/g DM). Studies on drying some red fruits with the same methods have reported higher TPC values in freeze-dried samples [15,30]. FD takes place in low temperature and vacuum circumstances, which explains the high TPC value. Investigated were the effects of various drying methods on the Tunisian eggplant's drying characteristics and some of its bioactive attributes. Similar to our study, this one demonstrated that freeze-drying and ultrasound-assisted drying caused the lowest product degradation [40]. Another study reported the benefits of ultrasonic treatment on the quality of the dried kiwi product. [41]. In comparison to hot-air drying, raspberry samples dried using ultrasound-assisted vacuum drying contained higher bioactive components, antioxidant properties, and improved color quality, according to Tekin Cakmak and Kayacan Cakmakoglu [15]. The formation of cavitation, which makes the components released from the cells during drying easier to extract, is associated to the high TPC value of goji berry samples dried with the USVD technique.

**Table 3.** Bioactive properties of fresh and dried goji berries.

| Properties | Fresh | Dried Goji Berry | | | | | r | |
| | | FD | USFD | VD | USVD | HAD | DPPH | CUPRAC |
|---|---|---|---|---|---|---|---|---|
| TPC [1] | 772.20 ± 9.31 [e] | 1115.03 ± 16.82 [b] | 1006.78 ± 5.17 [d] | 1060.54 ± 11.68 [c] | 1238.59 ± 66.19 [a] | 1002.54 ± 0.00 [d] | | |
| DPPH [2] | 15.70 ± 0.32 [d] | 24.82 ± 1.33 [bc] | 23.26 ± 0.68 [c] | 24.97 ± 1.77 [bc] | 28.19 ± 3.71 [ab] | 29.90 ± 2.04 [a] | 1 | |
| CUPRAC [3] | 40.98 ± 5.28 [d] | 203.39 ± 4.97 [b] | 152.98 ± 4.46 [c] | 210.35 ± 10.66 [ab] | 226.09 ± 10.00 [a] | 206.35 ± 16.91 [b] | | 1 |

FD: freeze-drying method; USFC: ultrasound-assisted freeze-drying method; HAD: hot-air drying method; USVD: ultrasound-assisted vacuum drying method; VD: vacuum drying method. The different lowercase letter in same line shows statistical significance ($p < 0.05$). [1] TPC: mg GAE/g DM; [2] DPPH: mg TE/g DM; [3] CUPRAC: mg TE/g DM.

By using the DPPH and CUPRAC techniques, the antioxidant capacity of fresh samples was found to be 15.70 mg TE/g DM and 40.98 mg TE/g DM, respectively. Similar to the TPC results, the highest DPPH and CUPRAC results were found to be, respectively, 28.19 mg TE/g DM and 226.09 mg TE/g DM, in goji berry samples dried using the USVD method. The USVD method is followed by the FD method (24.82 mg TE/g DM) as a result of DPPH, while the CUPRAC result is followed by the VD method (210.35 mg TE/g DM). The degradation of phenolic components and decreased antioxidant activity of the samples might have been brought about by thermal treatments and oxidative processes [42]. Antioxidant capacity of fruits dried with USVD method is higher than samples dried with HAD as in TPC results. These results show that the USVD method can be an alternative method to the HAD method. The Pearson coefficients (r) are shown in Table 3.

### 3.3. The Phenolic Profile of Fresh and Dried Goji Berries

Table 4 showed the effect of different drying methods on individual phenolic compounds. Protocatechuic acid (1.421 mg/g DM) and catechin (1.029 mg/g DM) were determined as the most prevalent phenolic substances in fresh goji berry samples. Catechin level were lower in all dried samples as they were sensitive to oxidation by the polyphenol oxidase enzyme, acid, and heat (Table 4). Gallic acid was identified as the main phenolic compound in the dried goji berry samples among the standards tested, followed by protocatechuic acid, rutin, catechin, and chlorogenic acid, although this differed from sample to sample. The effect of drying techniques on the distribution of phenolic compounds was found to be significant ($p < 0.05$). The HAD method resulted in a reduction in the amount of all identified phenolic compounds. While HAD caused a decrease in the amount of gallic acid, all other drying methods caused a serious increase. When the dried samples

were examined, Gallic acid, myricetin, and quercetin were the highest in the samples dried with FD; protocatechuic acid and caffeic acid were the highest in samples dried with VD; myricetin and quercetin were the highest in samples dried with UFD; and samples dried with USVD contained the highest values of catechin, chlorogenic acid, 4-hydroxybenzoic acid, ellagic acid, p-coumaric acid and ferulic acid. These results were in agreement with the findings of Donno, Mellano [43]. They reported that coumaric acid, ferulic acid, and catechin were the most abundant phenolic compounds. As a result, the effect of different drying methods on phenolic component distribution is consistent with total phenolic compounds and antioxidant activity results. While HAD caused a significant decrease, FD showed high results overall. The USVD method generally showed the highest amount of phenolic compounds. Terzić, Majkić [44] studied the influence of the different drying and extraction processes on the bioactive properties of the elderberry fruit. It was reported that extracts obtained from lyophilized fruit showed higher phenolic acid than conventional dried fruit extracts. Their results were consistent with our study. Donno, Mellano [43] studied the effect of air thermal drying and freeze-drying on the content of phytochemicals of goji berry fruits including the phenolic acids, catechins, and vitamins. Freeze dried samples showed higher phytochemicals than those of air thermal dried samples.

**Table 4.** Phenolic acid composition of goji berries.

| Phenolic Compounds (mg/g DM) | FRESH | FD | USFD | VD | USVD | HAD |
|---|---|---|---|---|---|---|
| Gallic acid | 0.160 [d] | 3.416 [a] | 2.811 [b] | 2.225 [c] | 2.860 [b] | 0.142 [d] |
| Protocatechuic acid | 1.421 [b] | 1.292 [c] | 1.253 [c] | 1.781 [a] | 1.753 [a] | 0.592 [d] |
| Catechin | 1.029 [a] | 0.341 [d] | 0.397 [c] | 0.302 [e] | 0.431 [b] | 0.234 [f] |
| Chlorogenic acid | 0.314 [a] | 0.151 [c] | 0.154 [c] | 0.126 [d] | 0.177 [b] | 0.031 [e] |
| 4-hydroxybenzoic acid | 0.150 [a] | 0.056 [c] | 0.149 [a] | 0.128 [b] | 0.153 [a] | 0.007 [d] |
| Syringic acid | 0.010 [a] | 0.004 [b] | 0.012 [a] | 0.013 [a] | 0.019 [a] | n.d. |
| Ellagic acid | 0.181 [a] | 0.099 [b] | 0.122 [b] | 0.106 [b] | 0.127 [b] | n.d. |
| Caffeic acid | 0.043 [a] | 0.051 [a] | 0.045 [a] | 0.061 [a] | 0.051 [a] | 0.050 [a] |
| p-coumaric acid | 0.211 [a] | 0.082 [b] | 0.076 [b] | 0.080 [b] | 0.084 [b] | 0.013 [c] |

FD: freeze-drying method; USFD: ultrasound-assisted freeze-drying method; HAD: hot-air drying method; USVD: ultrasound-assisted vacuum drying method; VD: vacuum drying method. The different lowercase letter in the same line shows statistical significance.

### 3.4. Color Parameters of Fresh and Dried Goji Berries

For dried products, color is the crucial quality factor that determines the acceptance of consumers of the final product. Table 5 displays the fresh and dried goji berry samples' color properties. $L^*$, $a^*$, $b^*$, and C values of fresh samples are 42.90, 29.7, 23.35, and 37.83, while $L^*$ values of dried samples range from 29.83–41.89, $a^*$ values range from 17.52–31.95, $b^*$ values range from 6.86–21.81, and C values range from 18.81–37.51.

**Table 5.** Color parameters of fresh and dried goji berries.

| Color Parameters | FRESH | Goji Berry | | | | |
| | | HAD | VD | USVD | FD | USFD |
|---|---|---|---|---|---|---|
| $L^*$ | 42.90 ± 3.08 [a] | 29.83 ± 1.71 [c] | 36.50 ± 0.94 [b] | 39.51 ± 0.22 [a] | 40.74 ± 2.24 [a] | 41.06 ± 1.06 [a] |
| $a^*$ | 29.77 ± 0.94 [a] | 17.52 ± 1.67 [b] | 29.52 ± 2.59 [a] | 27.65 ± 1.93 [a] | 31.95 ± 1.73 [a] | 28.41 ± 1.71 [a] |
| $b^*$ | 23.35 ± 1.54 [a] | 6.86 ± 1.15 [c] | 21.81 ± 0.81 [b] | 19.66 ± 1.75 [b] | 19.56 ± 0.69 [b] | 18.39 ± 1.92 [b] |
| ΔE | - | 23.93 | 5.80 | 4.97 | 4.50 | 5.24 |
| C | 37.83 [a] | 18.81 [c] | 36.70 [a] | 33.93 [b] | 37.51 [a] | 33.85 [b] |

FD: freeze-drying method; UDK: ultrasound-assisted freeze-drying method; HAD: hot-air drying method; USVD: ultrasound-assisted vacuum drying method; VD: vacuum drying method. Different lowercase letters in the same row indicate differences between samples subjected to different drying methods ($p < 0.05$).

The drying methods significantly affected the $L^*$ values of goji berry samples ($p < 0.05$). HAD causes a decrease in the $L^*$ value of the sample ($p > 0.05$). The decrease in $L^*$ values

could be caused by enzymatic and non-enzymatic browning mechanisms [45]. USVD has a faster drying rate and occurred under a vacuum. Therefore, the less non-enzymatic reaction could happen. Additionally, the pore structure is conserved in USVD, and this may have led to more reflection and less reduction in the *L\** value of the sample. The FD samples were dried under vacuum at low temperatures and the formation of the Maillard reaction product could be limited compared to thermal drying methods including conventional hot-air drying [46]. This explains the lesser decrease in *L\** value in samples dried with FD. Goji berries have carotenoids with a high content and the carotenoids are primarily responsible for red-orange goji berry fruits [47]. While the *a\** values of VD, USVD, FD, and USFD goji berries were unchanged, an important decrease was determined for HAD goji berries. This decrease could be explained by carotenoid degradation during HAD drying [30]. As in *L\** and *a\** values, there was a noticeable decline in the samples' *b\** and C values after they were dried using the HAD technique.

In a study, pomegranate samples were dried at 50 °C with HAD, VD, USVD and FD methods, and the highest decreases in the *L\** value occurred in samples dried using the HAD method [5]. The rosehip fruit, a red fruit similar to the goji berry and pomegranate, was dried by different methods on the color parameters and it was found that the sample dried with the HAD method had the lowest *L\** value, while the sample dried by the FD method had a higher *L\** value than the fresh samples [30]. The ΔE value is a critical indicator of the overall color change in dried samples. For dried goji berry samples, the highest ΔE value was found for the HAD method (23.93) and the lowest ΔE value was found for FD (4.50). Song, Bi [48] similarly reported the highest *L* value and lowest ΔE value were obtained from FD-dried samples. USVD showed lower ΔE value than those of VD and HAD. This result is in agreement with the Li, Wang [45] study. The lower ΔE value of the USVD compared to VD and HAD might be due to the lower drying time and the inhibitory effect of ultrasound application on browning enzymes especially polyphenol oxidase [49,50]. VD dried samples showed a lower ΔE value than that of HAD dried samples. This result is consistent with the finding of Orikasa, Koide [51]. While the C value of fresh goji berries was 37.83, this value was 18.81 after HAD. Minimal C value change was detected in FD samples. A high C value means that the intensity of color perceived by people is high. C values of VD and FD were found to be higher than USVD and USFD. It was stated that the application of ultrasound induced pigment degradation. The results of the ΔE and C values were consistent with the images of the samples.

## 4. Conclusions

In this study, fresh goji berries were dried with five different drying methods, and changes in drying kinetics, bioactive components, and color properties were investigated. The drying characteristic of the goji berry fruit was modeled employing nine different models. The logarithmic model showed higher $R^2$ and lower RMSE values for the HAD method, for the Parabolic VD method, and for the Page and Weibull USVD method than the other models. USVD showed the highest drying rate among the thermal treatments. This means that the combination of ultrasound and vacuum processes provided a shorter drying time than HAD and VD. USVD was found to preserve the total phenolic content the most in fresh goji berry samples. This method is followed by FD, VD, and HAD. High temperature and oxidative reactions in hot-air drying caused the loss of phenolic compounds. When all parameters are evaluated, the best alternative preservation method for drying goji berry fruit is freeze-drying, but FD has a long drying time and high costs. The bioactive components were less preserved in the samples of goji berry fruit dried by ultrasound pretreatment compared to VD and FD due to the waxy layer and low moisture content.

**Author Contributions:** Data curation: S.K.Ç., Z.H.T.-C. and B.T.; Conceptualization, S.K.; methodology, S.K.; writing—original draft, S.K.Ç., E.A. and Z.H.T.-C.; investigation, S.K., E.A. and S.K.Ç.; writing—review and editing, S.K. and M.Z.K.; visualization; Z.H.T.-C. and S.K. All authors have read and agreed to the published version of the manuscript.

**Funding:** This research received no external funding.

**Institutional Review Board Statement:** Not applicable.

**Informed Consent Statement:** Not applicable.

**Data Availability Statement:** The authors declare that all the data supporting the findings of this study are available within the article.

**Conflicts of Interest:** The authors declare no conflict of interest.

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
