# Peer review of "Effect of Different Drying Techniques on Total Bioactive Compounds and Individual Phenolic Composition in Goji Berries"

_processes, doi:10.3390/pr11030754_

Round 1
Reviewer 1 Report
The manuscript “Effect of different drying techniques on total bioactive compounds, phenolic, and carotenoids composition, and color change of goji berries” is effect of different drying methods namely, hot air drying (HAD), ultrasound-assisted vacuum drying (USVD), vacuum drying (VD), freeze-drying (FD), and ultrasound-assisted freeze drying (USFD) on the drying time and quality characteristics of goji berry. The manuscript will attract the readers. Some suggestions are given below to improve the readability of MS.
· Title
The title of a manuscript should be concise, precise and may represent all aspects of the study. Current title of the manuscript is quite long and may be reduced. The title may be “Effect of different drying techniques on total bioactive compounds, phenolic and carotenoids composition in goji berries”
· Abstract
Page 1, line # 19-20, “the total phenolic content TPC) value of dry samples varied” should be replaced as “the total phenolic content TPC) value of dry samples varied”.
Page 1, line# 22, what is “ΔE”? Please write complete name at first use.
Moreover, please add a concluding statement in the abstract along with an outlook. This will increase the readability of the manuscript.
· Introduction
Introduction of the MS is so important. There should be a reason why various methods of drying reducing the chemical properties of the goji berries. In introduction various drying methods may be compared for fruit drying, which is lacking currently.
· Material and Methods
In material and methods all drying methods are compared with each other. How quality parameters are maintained in Gogi barriers.
· Results and discussions
Results on drying kinetics, phenolics and colour is presented well but all the parameters discussion is lacking.
All the best!
Author Response
Comments and Suggestions for Authors
Reviewer 1
The manuscript “Effect of different drying techniques on total bioactive compounds, phenolic, and carotenoids composition, and color change of goji berries” is the effect of different drying methods namely, hot air drying (HAD), ultrasound-assisted vacuum drying (USVD), vacuum drying (VD), freeze-drying (FD), and ultrasound-assisted freeze drying (USFD) on the drying time and quality characteristics of goji berry. The manuscript will attract readers. Some suggestions are given below to improve the readability of MS.
- Title
Comment: The title of a manuscript should be concise, and precise and may represent all aspects of the study. Current title of the manuscript is quite long and may be reduced. The title may be “Effect of different drying techniques on total bioactive compounds, phenolic and carotenoids composition in goji berries”
Response: Thank you for your advice. The title of the manuscript was revised to “Effect of different drying techniques on total bioactive compounds and individual phenolic composition in goji berries”
Abstract
Comment: Page 1, line # 19-20, “the total phenolic content TPC) value of dry samples varied” should be replaced as “the total phenolic content (TPC) value of dry samples varied”.
Response: Line 1-21 was revised to “USVD treatments exhibited a higher total phenolic content (TPC) value than all other drying methods. DPPH and CUPRAC values varied between 15.70-29.90 mg TE/g DM and 40.98-226.09 mg TE/g DM, respectively.”
Comment: Page 1, line# 22, what is “ΔE”? Please write complete name at first use.
Response:: Thank you. We revised it. “The total color change (ΔE) value ranged between 4.59 and 23.93 and HAD dried samples showed the highest ΔE than that of other samples.”
Comment: Moreover, please add a concluding statement in the abstract along with an outlook. This will increase the readability of the manuscript.
Response: Thank you for your advice. We add this sentence. “The results of this study concluded that USVD and FD could be used as alternative methods to HAD in the drying of goji berries.”
- Introduction
Comment: The introduction of the MS is so important. There should be a reason why various methods of drying reduce the chemical properties of the goji berries. In the introduction, various drying methods may be compared for fruit drying, which is lacking currently.
Response: The introduction part was removed by new references. The background of the manuscript was explained. Application reason of the vacuum drying and ultrasound assisted vacuum drying was explained. The several drying methods were applied for fruit drying. It is seen that there are many methods in the literature for drying goji berry. The effects of these methods on the drying time, color and bioactive properties of goji berry fruit were investigated in detail. FD and HAD are drying techniques with different advantages and disadvantages. There is a need to propose alternative techniques to these techniques.
Material and Methods
Comment: In material and methods, all drying methods are compared with each other. How quality parameters are maintained in Goji berries.
Response: The lower degradation in bioactive compounds, lower drying times, and lower change in color value are considered indications of the success of the methods. The following sentences were added in the method section.
“The dried samples were compared with the fresh samples. The lower change in color and bioactive compounds was considered as a success criterion of the methods”.
- Results and discussions
Results on drying kinetics, phenolics, and color are presented well but all the parameters discussion is lacking.
Response: The discussion parts were improved following new cited references;
Drying kinetics:
Ni, Jiabao; Ding, Changjiang; Zhang, Yaming; Song, Zhiqing (2020). Impact of different pretreatment methods on drying characteristics and microstructure of goji berry under electrohydrodynamic (EHD) drying process. Innovative Food Science & Emerging Technologies, (), 102318–. doi:10.1016/j.ifset.2020.102318 .
Fernandes, F. A., Oliveira, F. I., & Rodrigues, S. (2008). Use of ultrasound for dehydration of papayas. Food and Bioprocess Technology, 1, 339-345.
Luis Puente-Díaz, Kong Ah-Hen, Antonio Vega-Gálvez, Roberto Lemus-Mondaca & Karina Di Scala (2013) Combined Infrared-Convective Drying of Murta (Ugni molinae Turcz) Berries: Kinetic Modeling and Quality Assessment, Drying Technology, 31:3, 329-338, DOI: 10.1080/07373937.2012.736113.
Phenolic compounds and color
X.-F. Cheng, M. Zhang, B. Adhikari, The inactivation kinetics of polyphenol oxidase in mushroom (Agaricus bisporus) during thermal and thermosonic treatments, Ultrason. Sonochem. 20 (2013) 674–
- Zhang, L. Niu, D. Li, C. Liu, R. Ma, J. Song, J. Zhao, Low intensity ultrasound as a pretreatment to drying of daylilies: Impact on enzyme inactivation, color changes and nutrition quality parameters, Ultrason. Sonochem. 36 (2017) 50–58.
Milena Terzić, Tatjana Majkić, Gökhan Zengin, Ivana Beara, Carlos L. Cespedes-Acuña, Dejan ÄŒavić, Marija Radojković. Could elderberry fruits processed by modern and conventional drying and extraction technology be considered a valuable source of health-promoting compounds?, Food Chemistry, 405, Part A, 2023, https://doi.org/10.1016/j.foodchem.2022.134766
Xiaomin Tang, Yaqiong Zhang, Feiyang Li, Na Zhang, Xiaoyu Yin, Bo Zhang, Bolin Zhang, Wenrui Ni, Mengze Wang, Junfeng Fan. Effects of traditional and advanced drying techniques on the physicochemical properties of Lycium barbarum L. polysaccharides and the formation of Maillard reaction products in its dried berries, Food Chemistry, 409,2023, https://doi.org/10.1016/j.foodchem.2022.135268
Huihui Song, Jinfeng Bi, Qinqin Chen, Mo Zhou, Xinye Wu & Jianxin Song (2018) Structural and health functionality of dried goji berries as affected by coupled dewaxing pre- treatment and hybrid drying methods, International Journal of Food Properties, 21:1, 2527-2538, http://doi.org/10.1080/10942912.2018.1536148
Donno, D., Mellano, M.G., Raimondo, E. et al. Influence of applied drying methods on phytochemical composition in fresh and dried goji fruits by HPLC fingerprint. Eur Food Res Technol 242, 1961–1974 (2016). https://doi.org/10.1007/s00217-016-2695-z.
- Orikasa, S. Koide, S. Okamoto, T. Imaizumi, Y. Muramatsu, J.-I. Takeda, T. Shiina, A. Tagawa, Impacts of hot air and vacuum drying on the quality attributes of kiwifruit slices, J. Food Eng. 125 (2014) 51–58

Reviewer 2 Report
Article ID: processes-2158885
Title: Effect of different drying techniques on total bioactive compounds, phenolic, and carotenoids composition, and color change of goji berries
This study aims to investigate the effect of different drying methods: hot air drying (HAD), ultrasound-assisted vacuum drying (USVD), vacuum drying (VD), freeze-drying (FD), and ultrasound-assisted freeze drying (USFD) on the drying time and quality characteristics of goji berry.
The current issue is interesting and worth investigating; however, for the manuscript to be ready for publication, some areas of opportunity must be addressed. Therefore, here are some comments that should be answered.
- Could you specify how long the ultrasound process last?
- Could you describe how the Ultrasound Assisted Freeze- Drying process was?
- Could you mention why it is essential to establish different drying models for the investigated technologies (HAD, VD, and USVD)?
- Compare the Deff values obtained for goji berries with other berries in the literature.
- It is unclear why the FD and USFD technologies were added to investigate the quality characteristics of goji berries.
- Table 5 is missing.
- The introduction and result discussion should be enriched with the following references.
Reference
Milena Terzić, Tatjana Majkić, Gökhan Zengin, Ivana Beara, Carlos L. Cespedes-Acuña, Dejan ÄŒavić, Marija Radojković. Could elderberry fruits processed by modern and conventional drying and extraction technology be considered a valuable source of health-promoting compounds?, Food Chemistry, 405, Part A, 2023, https://doi.org/10.1016/j.foodchem.2022.134766
Xiaomin Tang, Yaqiong Zhang, Feiyang Li, Na Zhang, Xiaoyu Yin, Bo Zhang, Bolin Zhang, Wenrui Ni, Mengze Wang, Junfeng Fan. Effects of traditional and advanced drying techniques on the physicochemical properties of Lycium barbarum L. polysaccharides and the formation of Maillard reaction products in its dried berries, Food Chemistry, 409,2023, https://doi.org/10.1016/j.foodchem.2022.135268
Huihui Song, Jinfeng Bi, Qinqin Chen, Mo Zhou, Xinye Wu & Jianxin Song (2018) Structural and health functionality of dried goji berries as affected by coupled dewaxing pre- treatment and hybrid drying methods, International Journal of Food Properties, 21:1, 2527-2538, http://doi.org/10.1080/10942912.2018.1536148
Donno, D., Mellano, M.G., Raimondo, E. et al. Influence of applied drying methods on phytochemical composition in fresh and dried goji fruits by HPLC fingerprint. Eur Food Res Technol 242, 1961–1974 (2016). https://doi.org/10.1007/s00217-016-2695-z
Author Response
Comments and Suggestions for Authors
Reviewer 2
Title: Effect of different drying techniques on total bioactive compounds, phenolic, and carotenoids composition, and color change of goji berries
This study aims to investigate the effect of different drying methods: hot air drying (HAD), ultrasound-assisted vacuum drying (USVD), vacuum drying (VD), freeze-drying (FD), and ultrasound-assisted freeze drying (USFD) on the drying time and quality characteristics of goji berry.
The current issue is interesting and worth investigating; however, for the manuscript to be ready for publication, some areas of opportunity must be addressed. Therefore, here are some comments that should be answered.
- Could you specify how long the ultrasound process last?
Answer: For the USVD method, goji berries were held at 30 min ultrasound water bath (Daihan, WUC-D10H, South Korea) (amplitude: 100%, power intensity: ~1 W/cm2, volume: 10 L) and then dried in a vacuum drier (DaihanWOV-30, Gangwondo, South Korea).
- Could you describe how the Ultrasound Assisted Freeze- Drying process was?
Answer: For the the Ultrasound Assisted Freeze- Drying (USFD) method, goji berries were held at 30 min ultrasound water bath (Daihan, WUC-D10H, South Korea) (amplitude: 100%, power intensity: ~1 W/cm2, volume: 10 L) and then dried in freeze dryer (Martin Christ, Beta 1–8 LSC plus), and the samples were frozen at -80 °C for 24h. The ultrasound assisted was changed as ultrasound pretreated Freeze- Drying (USFD).
- Could you mention why it is essential to establish different drying models for the investigated technologies (HAD, VD, and USVD)?
Answer: Following sentences were added to related part,
The thin layer drying modeling is useful for optimizing drying conditions, developing and optimizing dryers, and researching complex heat and mass transport processes (Akter et al, 2022). For this reason, It is essential to determine the most suitable model representing the drying behavior of the foods
- Compare the Deff values obtained for goji berries with other berries in the literature.
Answer: The new references were added to comparisons of the Deff values.
- It is unclear why the FD and USFD technologies were added to investigate the quality characteristics of goji berries.
Answer: As mentioned in the introduction, FD method is considered to be the best drying method due to its advantages such as high rehydration capacity and no change in color and bioactive properties of foods dried with FD. other drying methods were compared with FD. FD also has some disadvantages. The most important disadvantage is that the drying time is long and the cost is high. In the second part of our study, the effect of ultrasound pretreatment on the reduction of drying time will be studied. In other words, drying kinetics study will be done in FD. In this study, the properties of ultrasound pre-treaded samples before FD were also examined in order to see the possible effects of ultrasound treatment on the FD dried samples. If you suggest, we can remove the data of USFD from this study.
- Table 5 is missing.
Answer: Table 5 was added in manuscript.
- The introduction and result discussion should be enriched with the following references.
Answer: Thank you very much for your suggestions. All references listed below were cited for the improvement of different discussion parts.
Reference
Milena Terzić, Tatjana Majkić, Gökhan Zengin, Ivana Beara, Carlos L. Cespedes-Acuña, Dejan ÄŒavić, Marija Radojković. Could elderberry fruits processed by modern and conventional drying and extraction technology be considered a valuable source of health-promoting compounds?, Food Chemistry, 405, Part A, 2023, https://doi.org/10.1016/j.foodchem.2022.134766
Xiaomin Tang, Yaqiong Zhang, Feiyang Li, Na Zhang, Xiaoyu Yin, Bo Zhang, Bolin Zhang, Wenrui Ni, Mengze Wang, Junfeng Fan. Effects of traditional and advanced drying techniques on the physicochemical properties of Lycium barbarum L. polysaccharides and the formation of Maillard reaction products in its dried berries, Food Chemistry, 409,2023, https://doi.org/10.1016/j.foodchem.2022.135268
Huihui Song, Jinfeng Bi, Qinqin Chen, Mo Zhou, Xinye Wu & Jianxin Song (2018) Structural and health functionality of dried goji berries as affected by coupled dewaxing pre- treatment and hybrid drying methods, International Journal of Food Properties, 21:1, 2527-2538, http://doi.org/10.1080/10942912.2018.1536148
Donno, D., Mellano, M.G., Raimondo, E. et al. Influence of applied drying methods on phytochemical composition in fresh and dried goji fruits by HPLC fingerprint. Eur Food Res Technol 242, 1961–1974 (2016). https://doi.org/10.1007/s00217-016-2695-z.
Ferdusee Akter, Ripa Muhury, Afroza Sultana, and Ujjwal Kumar Deb. A Comprehensive Review of Mathematical Modeling for Drying Processes of Fruits and Vegetables. Volume 2022 | Article ID 6195257 | https://doi.org/10.1155/2022/6195257.

Reviewer 3 Report
line 54-55 "Thanks to ultrasonic waves, microscopic cavities are formed in the tis such a level of the article and writing about the kinetics of drying - it is necessary to use specialist terms such as free water, bound water. Please refer to these terms in this sentence.
The background is not entirely consistent with the purpose of the work. In the introduction, the authors mention only various drying techniques, omitting the topic of activities preceding the drying process and shaping its course - such as blanching or osmotic dehydration. Please complete the.
line 84 "Goji berries were dried by HAD, VD, USVD, FD, and USFD." - when discussing for the first time, it is necessary to clearly expand the abbreviations.
in methodology - please pay more attention to the correct use of subscripts
2.2.3.Extraction procedure - from the description of this method can you guess that the fruit was whole?
fig 1 - what is the purpose of presenting the same photo (a) five times?
3.1. Drying Kinetics - A bit of a poor discussion with only three science items
Table 3 Bioactive properties of fresh and dried goji berries - necessary correction, shifting of columns
3.4. Color parameters of fresh and dried goji berries - please present the results in the table
Author Response
Comments and Suggestions for Authors
Reviewer 3
Comments: line 54-55 "Thanks to ultrasonic waves, microscopic cavities are formed in the tis such a level of the article and writing about the kinetics of drying - it is necessary to use specialist terms such as free water, bound water. Please refer to these terms in this sentence.
Response: “Thanks to the ability of the ultrasound waves to squeeze and release the water repeatedly, microscopic cavities are formed in the tissues of fruits and vegetables, so that the bound water attached to the tissues and free water could be removed.”
Comments The background is not entirely consistent with the purpose of the work. In the introduction, the authors mention only various drying techniques, omitting the topic of activities preceding the drying process and shaping its course - such as blanching or osmotic dehydration. Please complete the.
Response :
Comments: line 84 "Goji berries were dried by HAD, VD, USVD, FD, and USFD." - when discussing for the first time, it is necessary to clearly expand the abbreviations.
Response: In the abstract, we clearly expand these abbreviations (Line 12-15).
“This study aims to investigate the effect of different drying methods namely, hot air drying (HAD), ultrasound-assisted vacuum drying (USVD), vacuum drying (VD), freeze-drying (FD), and ultrasound-assisted freeze drying (USFD) on the drying time and quality characteristics of goji berry.”
Comments : in methodology - please pay more attention to the correct use of subscripts
Response: Thank you very much for your attention. Eq 3 was revised based on the reviewer recommendation.
Comments : 2.2.3. Extraction procedure - from the description of this method can you guess that the fruit was whole?
Response: The dried and fresh fruit were macerated with solvent and ultra turraks was applied to grind and homogenise. The fruit were crushed and homogenized before the extraction.
Comments: fig 1 - what is the purpose of presenting the same photo (a) five times?
Response: We designed Figure 1 in this way to better see the change of each fresh sample after drying.
Comments : 3.1. Drying Kinetics - A bit of a poor discussion with only three science items
Response: The discussion part was improved by new references.
Comments : Table 3 Bioactive properties of fresh and dried goji berries - necessary correction, shifting of columns
Response: Necessary adjustments have been made for Table 3.
Comments : 3.4. Color parameters of fresh and dried goji berries - please present the results in the table
Response: Table 5 was added in the manuscript.

Round 2
Reviewer 1 Report
-
Reviewer 2 Report
Thank you for your answers